## Research Article

climate change; flood risk; household resilience; informal settlements; slum hazard; vulnerability

**Corresponding author:**
Sheku Gbetuwa;
Emails: shekugbetuwa93@gmail.com;
sheku.gbetuwa@tu-dortmund.de

# Households' readiness and community-based organisations' role in flood management: The case of Freetown City's coastal area

Bashiru Turay[1] , Sheku Gbetuwa[2] and Alieu Turay[3]

[1]Department of Geography, University of Bonn, Germany; [2]Faculty of Spatial Planning, Technische Universität Dortmund, Dortmund, Germany and [3]Department of Computer Science, University of Management and Technology, Freetown, Sierra Leone, West Africa

## Abstract

Flooding is a well-known extreme climate event affecting coastal settlements around the world. It is the principal climate-related disaster encountered by residents of Portee and Rukupa, coastal slums in Freetown, Sierra Leone. The impacts of floods in these slums have been exacerbated by the lack of effective control measures to address the disaster. One reason for this ineffectiveness is a lack of information about how households are ready to manage floods and the roles of community-based organisations (CBOs) in these events. Given this concern, this study examines household readiness and CBOs' roles in flood management in Portee and Rokupa using observation, purposive, and snowball sampling techniques to study 204 households and 12 CBOs. The results show that flood-related information in the community is mostly shared verbally among residents. In addition, most households claimed not to have received support amidst flood events, whereas CBOs within the area claimed the opposite. As such, we recommend that future studies look at household perceptions of vulnerability and willingness to take risk-reduction actions. This study encourages community members to strengthen inter-community and organisational learning, feedback, and support systems and adopt a "no wait on the government principle" for flood management.

## Impact statement

Flooding has been among the considerable safety and development concerns around the globe in recent decades. It is the most extreme climate event affecting the coastal settlements of Portee and Rokupa, Freetown City, Sierra Leone. Updated household socio-demographic information, including how ready residents are to manage recurring floods and the roles of community-based organisations (CBOs) in such events, is essential but previously missing. This situation leaves loopholes deterring the efficiency of local flood management and resulting in serious flood consequences. This study shows the reality on the ground through first-hand information sources regarding household access to flood information, sources of humanitarian aid, and community-based responses, which are vital to guiding the identification of effective measures to take for improving the status quo. The findings of this study are impactful and can be beneficial in aiding the government, international agencies, and CBOs in policymaking and project implementation necessary for achieving local climate management and sustainable development targets. The results can further serve as background knowledge and a guiding tool for future research concerning the understanding and addressing of issues in other coastal communities topographically resembling and facing similar socio-economic situations as the case of Portee and Rokupa.

## Introduction

Freetown City, Sierra Leone has been termed susceptible to flooding (GOPA-CES, 2014; Freetown City Council, 2022). Various flood events in the city have destroyed lives, properties, and livelihoods over the years. During these events, coastal slums, especially Portee and Rokupa are mostly affected. Their exposure to flooding is due to their low-lying geographical positions. The lower elevation puts these communities as recipients of both urban and coastal floods, which often result in disasters (YCARE International, 2012; GOPA-CES, 2014). Flooding in these communities follows a seasonal pattern, occurring every rainy season. Notable incidents include the August 25, 2020 flooding, which left about 60 people homeless and resulted in one fatality (Politico-SL, 2020). Another one is the September 2015 flood event which left about 27 homes affected and turned the roads in Portee and Rokupa into swift-moving rivers of water. Nine homes sustained extensive damage, and three others were destroyed. Two fatalities, some minor wounds, and some fractures were recorded (Macarthy et al., 2017).

The expansion of densely populated slums, poor waste management, and lack of adequate drainage systems are all factors that previous researchers have linked to flood disasters in coastal communities (YCARE International, 2012; Reingold, 2021; Turay and Gbetuwa, 2022). Over 68 areas in Freetown have been identified as slums or informal settlements (Oviedo et al., 2021). Despite the risks and dangers, the locals are hesitant to leave these communities mostly because of the ties to their sociocultural traditions and livelihood activities (Reingold, 2021).

People living in at-risk areas have been linked to unplanned urbanisation as the main factor. Freetown's continued urbanisation can be attributed in part to increased emigration from the interior. The majority of rural-to-urban migrants live in dangerous areas where they may find affordable housing, commonly in areas where habitation in an urban setting is prohibited (Ziervogel, 2021). Finding solutions to the aforementioned issues appears challenging because people move to urban areas in search of improved living conditions but are unable to make long-term plans for adaptation in their households and communities. Planning for long-term climate risks can be challenging due to uneven access to social amenities, the reality of living on a tight budget, and the difficulty in meeting basic needs (Satterthwaite et al., 2020).

Evidence has shown that other places in developing regions are facing similar challenges as Portee and Rokupa. For instance, according to research conducted in the coastal slum communities of Akoko, Ilaje-Bariga, Ijora-Oloye, and Marine Beach-Apapa in Lagos, people there lack access to basic infrastructure and services like clean water, electricity, stormwater drainage, roads, solid waste disposal, and sanitary conditions, as well as quality housing, and live in environmentally degraded situations. The slum areas' road networks are not well-maintained (Adelekan, 2010). Another study on urban slum inhabitants in the Indian Himalayan city of Dehradun found that their general vulnerability was very high and that they had a very limited capacity to absorb and adapt to anticipated climate change consequences. Furthermore, vulnerability and coping mechanisms differed socially depending on households' ability to make decisions and access resources (Pandey et al., 2018).

Household readiness, construed here as the state of planning and preparation of a housing unit for a current or expected harmful circumstance, is fundamental to hazard and disaster management. When this readiness is deemed insufficient or it is feared that it will not be sufficient to overcome the anticipated danger, household members organise or join existing social bodies in the community to benefit from the exchange of skills and ideas for the common good. These community-based organisations (CBOs) are bound by rules and committed to objectives established formally or informally. Through these organisations, informal and slum households are known to act as volunteers and first responders in addressing their immediate concerns brought on by increasing climate hazards and poor urban governance (Amoako and Inkoom, 2018). Though instrumental, findings from those in Freetown and Accra have shown that most of these organisations are incapacitated and unprepared to independently shoulder the impacts of flooding, probably due to a lack of support and recognition from public and non-governmental organisations and the high levels of economic vulnerability that characterise such locations (YCARE International, 2012; Macarthy et al., 2017; Amoako and Inkoom, 2018).

Macarthy and his colleagues previously investigated the relationships between empowerment and humanitarian responses in Portee and Rokupa using Amartya Sen's Capability Approach. They found that the most successful method for raising the empowerment assets of residents in Portee and Rokupa had been community-based humanitarian practices (Macarthy et al., 2017). Reingold also viewed the Portee and Rokupa cases, noting that CBOs would be better able to address issues relating to biodiversity and subsistence if they could coordinate humanitarian efforts with regional, national, and international government agencies and NGOs (Reingold, 2021).

There is no evidence showing the finalisation and implementation of the drafted Freetown structural plan 2013–2028 which was designed to provide a solution to most of the problems in the study area and other slum communities in Freetown. In addition, no study has been done to understand households' readiness and CBOs' response to flooding in the coastal area of Portee and Rokupa. Such information is essential to support the urban risk reduction and climate resilience policies and actions of the government and its partners. Given this background, this study aims to investigate household readiness and CBOs' responses to flooding.

This work is developed from community resilience as a theory and strategy for disaster readiness, established by (Norris et al., 2008). The theory depicts the complex relationship among community members as they strive to mobilise local resources using social supports and information, working through members' capacities and competencies in a joint effort to manage harm brought on by the disaster.

The following section details the methodology used, followed by a description of the case study, the results, and discussion sections. The study's conclusion section wraps up with the main points and next actions recommended for research and implementation.

## Methodology

This study used purposive and snowball sampling to survey household heads and CBOs' key informants. The authors complemented these techniques with observation due to its relevance to examining the physical conditions and other characteristics necessary to understanding households' vulnerability and exposure to flooding in the study area.

Previous reports showed an estimated 1,165.2 households in the study area (Macarthy et al., 2017; Statistics Sierra Leone, 2019; Turay, 2022). Conditioned by the scope of this study, the authors purposefully selected 204 of these for investigation. The selection of households for interviewing was based on their fitting into the criteria of a slum settlement, according to the UN, which describes them as a settlement that lacks one or more of the following criteria: improved sanitation, adequate living space, secure tenure, a durable housing structure, and overcrowded (UN-Habitat, 2018).

Twelve representatives of CBOs were also interviewed using a snowball sampling technique. Due to the complex social relationships and interactions in a community, members know one another (Bott et al., 2020; Paterson and Charles, 2019). This idea, coupled with the little information established by previous studies about CBOs and their roles, made snowballing the most suitable technique for contacting and interviewing the relevant organisational representatives.

An in-person mobile survey was used using the Kobo Toolbox for data collection. The Kobo Toolbox is a free and open-source tool for gathering field data in humanitarian settings and difficult environments. In contrast to paper-based questionnaires, Google forms, and other field data collection tools, the Kobo toolbox can collect data without an internet connection, record GPS coordinates, and is equipped with functions for descriptive statistics, data analysis, and visualisation; raw data can be exported into

Excel, SPSS, or QGIS/ArcGIS. The Kobo Toolbox has developers based in Cambridge, Massachusetts, United States of Americaand other parts of the world, with the United Nations High Commissioner for Refugees, the United Nations Office for the Coordination of Humanitarian Affairs, the Havard Humanitarian Initiative, the World Bank, etc., as sponsors. The validity and reliability of this tool has been shown by its increasing use by more than 14,000 non-governmental and international organisations, including all United Nations agencies, the International Red Cross, and the Red Crescent Movement, as well as individual researchers, such as (Beretta, 2021). The tool has extensively been used to conduct fieldwork in cases bearing a similar context to this study (Kobotoolbox, 2023). The questions contained in the questionnaire are consistent with the corresponding sub-sections of the results section.

Data were collected between April 7 and May 28, 2022. The household data collection exercise in Portee covered Kabba Street, Limba Wharf, and Munjuru Street. The locations covered in Rokupa are Rokupa Wharf, Wright Street, and the lower side of West Street. Where the household heads were not present at the time of the visit, the authors had to obtain consent from an adult household representative on their behalf. A revisit was performed before the end of the day's work in cases where no adult was present in a household. Throughout the data collection process, the assistance of a community-based expert was used. Following confidentiality and other research ethics, the authors ensured that all participants' names and personally identifiable information were anonymised and that they understood the purpose and methods of the study and freely consented to participate.

### Data analysis

Geospatial data to understand and illustrate the case study were analyzed in QGIS Desktop. Shapefile data containing Sierra Leone administrative boundaries were downloaded from https://data.humdata.org/dataset/cod-ab-sle. The boundary shapefile was clipped to the extent of the case study using the clip vector geoprocessing function. The hybrid topographical features were extracted from Google Hybrid and laid out using the QuickMap Services plugin. Likewise, elevation data were digitised and exported from Google Earth Pro, processed, and laid using the contour tool. For a comprehensive geospatial representation, various labeling techniques were also applied.

Data from the field were exported in Excel format from the Kobo Toolbox user account created for this study. In Excel, data for households and CBOs were collated, checked for consistency, cleaned, formatted as a table, and analyzed. Verbal responses were recorded on a smartphone, manually transcribed, and presented in the respective results section.

### The case study description

The three administrative regions of Freetown are Central, East, and West. These regions are further divided into sub-regions, that is, Central (i, ii), East (i, ii, iii), and West (i, ii, iii). The communities of interest, Portee and Rokupa, are in Freetown's East iii administrative sub-region. Portee is bound to the north by Grassfield, to the west by Kuntolor, and to the south by Rokupa. Rokupa is bordered by Portee on the north, Kuntolor on the west, and Congo Water on the south. The two communities are bounded by the Sierra Leone River, an estuary of the Atlantic Ocean to the east. A cliff-like feature stretches from the north to the south of the study areas

and separates the coastal settlements from the upper part. Figure 1 below shows the study area and surroundings.

Wards are Freetown's smallest political administration units, governed by ward councillors elected to the Freetown City Council (FCC) every 5 years. Given this context, the majority of the study area (the entire Portee and portion of Rokupa) falls within Ward 409, whereas the remaining portion of Rokupa is in Ward 408. The FCC is the city government body that is led by a mayor. (NEC-SL, 2017). Portee and Rokupa are home to 34,502 people, making up 3.4% of Freetown City's total population. This population resides in a densely populated area with low literacy and a high rate of poverty (NEC-SL, 2017; Statistics Sierra Leone, 2019). The main socio-economic activity is fishing and the processing and marketing of fisheries (Macarthy et al., 2017; Turay, 2022).

Before local government and decentralisation were restored in Sierra Leone in 2004, the two communities coexisted as one settlement. Due to the border demarcation for the first local elections in 2004, the communities were formally divided (Local Government Act, 2004).

Although it is more temperate inland, the climate is hotter and more humid near the coast. According to World Bank data (1990–2020), the wettest months in Freetown are June, July, and August, which have the highest average monthly rainfall of 1887.62 mm. The risk of flood disasters rises when there is a lot of rain because the Atlantic Ocean's water level rises and overflows, adding to the inland water that overflows from rivers, streams, and drainages that drain into coastal areas. Flooding in Portee and Rokupa is significantly influenced by rainfall as a result of climate change (World Bank, 2021).

Portee and Rokupa have been identified as flooding hotspots in earlier studies (GOPA-CES, 2014; World Bank, 2017). Portee is inhabited in places with elevations ranging from 0 to 58 meters, and Rokupa in places with elevations ranging from 0 to 50 meters above sea level. The coastal area (Figure 2), which ranges from about 0 to 10 meters, is the area most exposed to flooding and houses observed in slum conditions and was thus chosen for this study.

## Results

### Households characteristics

The authors took records of household sizes, age, marital status, the gender distribution of household heads, and length of residence to understand the connection between those characteristics and the readiness to manage floods. The results are illustrated in Table 1.

Gender is implied here as the socially constructed roles, behaviours, and identities of boys, girls, men, women, and other diverse groups. As shown in the table above, the majority (112) of household representatives surveyed are women and men (92) are the least. In terms of age, the mean age of the interviewees was 39.8 years. It can be deduced that a person's physical capabilities and responsiveness to emergencies are directly related to their age. The authors consider the mean age of respondents ideal for any flood management activity, though it could not ensure actions in solitude, as other factors such as education, skills and resources can also play important parts.

Moreover, more than half of the respondents (122) were married, followed by 36 singles, 32 widows or widowers, 11 divorcees, and 3 respondents who were separated. Being in a marital union can be associated with a shared perspective in protecting the home. However, the majority of married people in Portee and Rokupa did not show any apparent signs of flood protection.

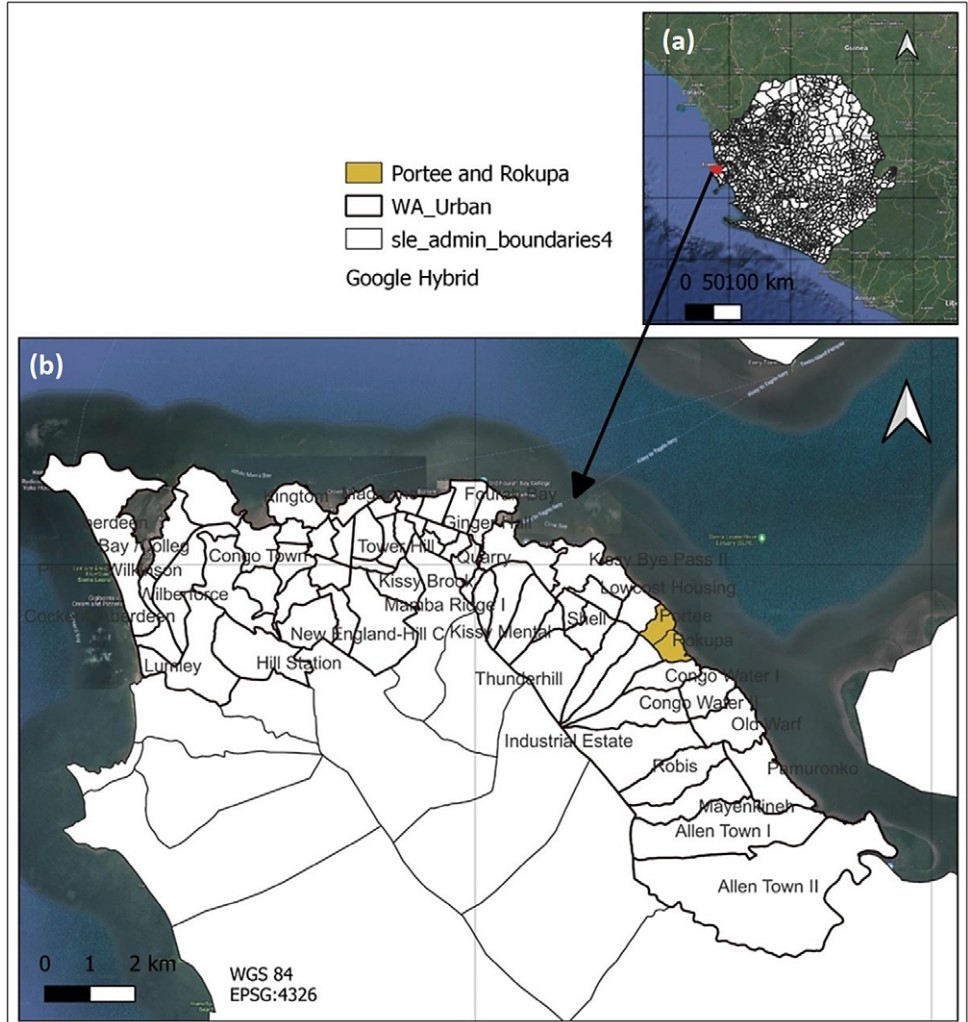

**Figure 1.** Sierra Leone (a), Western Area, Portee and Rokupa (b). Data source: https://data.humdata.org/dataset/cod-ab-sle, and Google hybrid layer. Accessed 20/1/2023. Drawn: Authors

Another key household characteristic observed in the study was the household size. In this aspect, 44 of the household interviews comprised four members, 40 of them have six members, 37 households have five members, and 27 have three members. Furthermore, 23 households claimed two members each, nine households claimed seven members each, 14 households had more than 7 members each, and 10 households claimed to be alone in their house. It can be deduced that a household with more members and insufficient resources may be more vulnerable due to the demand for those resources. The capacity of the entire household to respond to flooding, on the other hand, can be improved by a bigger household size with strong social networks and resources. Observations reveal that the 14 households with seven or more occupants reside in congested places and are thus more vulnerable to the negative effects of floods, such as the quicker spread of contagious waterborne illnesses (like cholera) brought on by the ingestion of flood-polluted water.

The findings further showed that 144 of the respondents had lived in their current residences for more than 5 years, whereas only 18 of those who had been interviewed had done so for between 3 and 4 years. Less than 3 years have passed since the remaining people moved into their homes. These individuals' duration of stays in flood-prone places may be related to both their prior flooding experiences and their inability or unwillingness to keep away from these risky areas.

### Flood information, readiness to respond and evacuate

Information about a potential flood event is vital for preparation and evacuation, where necessary. The authors, therefore, inquired about the interviewees' means of knowing about flooding. The results show that most (189) of the interviewees get information about floods by word of mouth. Although 65 of them said that they get such information through radios, phone calls, SMS, etc. Only 52 of the interviewees reported that they get flood information through social media communication channels, such as WhatsApp, Facebook, Twitter, etc. The remainder stated that they are aware of floods when they or their neighbours are affected. Most interviewees getting flooding information by word of mouth in Portee and Rokupa shows a widespread lack of flood information among them and a limited amount of time to prepare to address flooding.

The fieldwork was done during the rainy season, which is also when flood occurrences are likely to take place. In light of this,

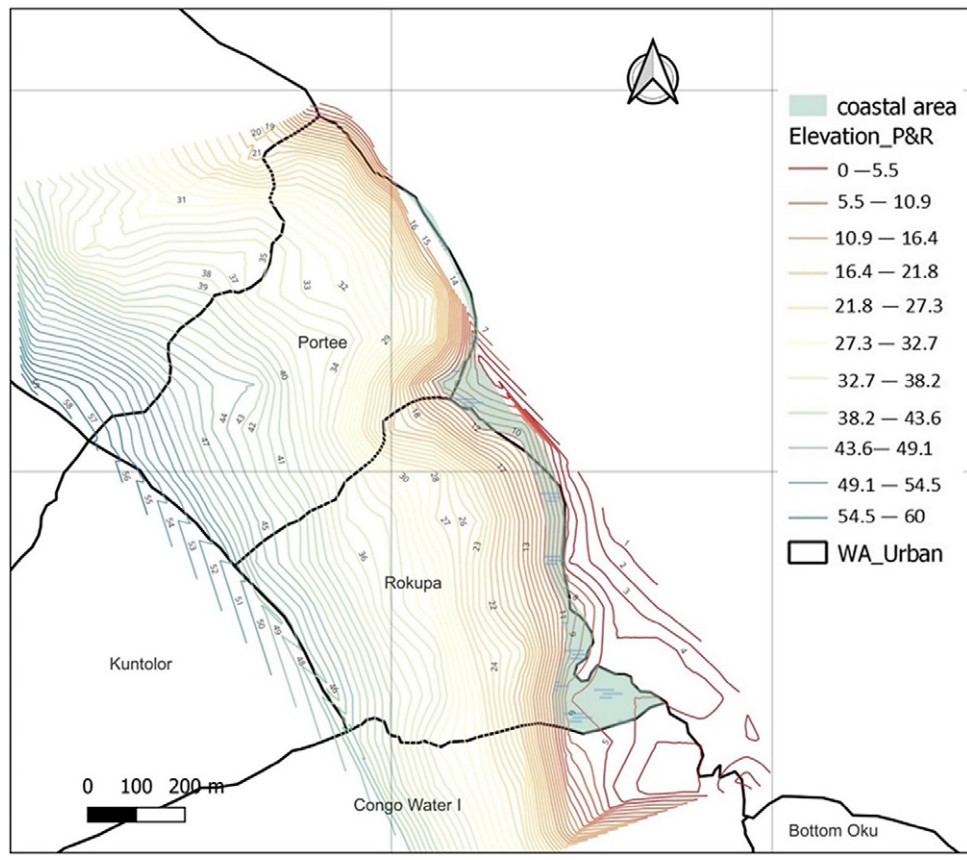

**Figure 2.** Elevation of the study site. Boundary shapefile, https://data.humdata.org/dataset/cod-ab-sle; elevation and imagery layers, Google earth pro and Google hybrid. Accessed: 12.08.2022. Drawn: Authors.

**Table 1.** Summary of household characteristics of respondents

| Gender | | Length of stay in the community | |
| --- | --- | --- | --- |
| Variable | Frequency | Variable | Frequency |
| Male | 92 | Less than 3 years | 42 |
| Female | 112 | 3–4 years | 18 |
| Total | 204 | 5 years and above | 144 |
| | | Total | 204 |

| Household size | | Marital Status | |
| --- | --- | --- | --- |
| Variable | Frequency | Variable | Frequency |
| 1 | 10 | Single | 36 |
| 2 | 23 | Married | 112 |
| 3 | 27 | Widowed | 32 |
| 4 | 44 | Divorced | 11 |
| 5 | 37 | Separated | 3 |
| 6 | 40 | Total | 204 |
| 7 | 9 | | |
| 7+ | 14 | | |
| Total | 204 | | |

respondents were questioned about whether they were ready to evacuate to a safer location in the event of a flood. The majority (122) of them said they did not think they were ready; 46 said they were doubtful of readiness; and the remaining 36 said they were ready to go if necessary. Further questions focused on whether interviewees had a temporary safe location they could go to during an evacuation. A temporary haven is available for 31 respondents, whereas 83 said their households have nowhere to go in case of need. The fact that the majority of them do not believe their households are ready to flee to safety (even if the need arises) suggests there is a chance they could become stuck in floods and experience the negative effects that go along with it. Additionally, the availability, accessibility, and distance of a suitable temporary evacuation area, as well as cultural, religious or livelihood considerations, may be related to the majority of interviewees stating that they lack a place to evacuate in an emergency.

The number of vulnerable people that households need to be concerned about in the event of an evacuation was also questioned. According to the authors, those who are at least 65 years old and those younger than 10 years old, as well as those with hearing or vision impairments, pregnant women, mental health issues, regular seizure sufferers, and mobility-impaired individuals are all mostly at risk during an evacuation. In terms of concern towards these people during an evacuation, 157 respondents reported to be concerned about them, whereas the remaining 47 respondents stated otherwise.

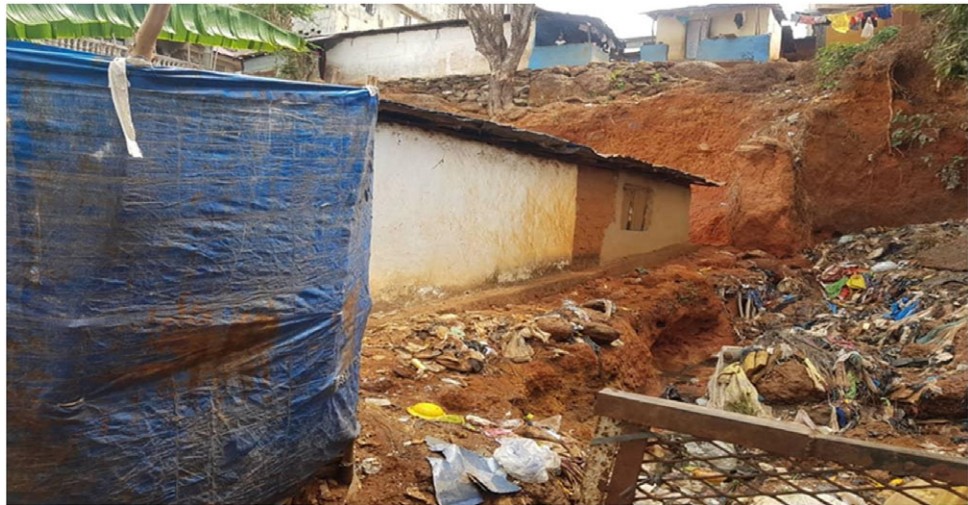

**Figure 3.** An example of a building exposed to rockfall and mudslide in the event of a heavy downpour and consequential higher soil infiltration. Source: Fieldwork.

In a separate question, a household head was asked how ready he is to handle flooding. "As for flooding, I am aware that it occurs every year during the rainy season, and I have my shovel ready to open more waterways in my gutter if there is a need. What I am more worried about is falling rock or mud, which is not frequent but has been happening." He responded. In the event of a heavy downpour and the resulting high infiltration, it was observed that chances of rock fall and mudslides are likely for houses built underneath the cliff-like structure of the area. Poorly constructed houses are exposed to serious consequences. An example of such a house is given in Figure 3.

### Source of humanitarian aid

Having someone to rely on for assistance is critical for recovery and provides emotional relief in distress. The availability of aid can be considered an important factor in flood preparation and management. With this perspective, the authors inquired to know from where households can rely on support when in need. The results show that the majority (173) of those interviewed do not consider their households to have any reliable humanitarian aid source. Only one considered CBOs and relatives as reliable sources of aid. 22 considered government/ public institutions a reliable humanitarian aid source, whereas 10 of the interviewees considered both local and international non-governmental organisations a reliable source of aid in the event of flooding. The interviewees' responses were based on their past experiences of support received trust and feelings of reliability about the sources of support mentioned.

### Community-based organisations' role in flood management

The authors investigated CBO's role in flood management. Of the 12 organisations investigated in the study, nine of them stated to have engaged in public sensitisation to increase community awareness in flood response and all of them are involved in clearing drainages and installing sandbags and other flood barriers during flooding. Only one CBO claimed to be active in investigating flood problems. Although several of these organisations' principal responsibilities do not involve flood management, they emphasised

that to protect their communities, they must participate. Table 2 is a summary of the reports provided by the CBOs interviewed.

Regarding the source of support provided to these organisations in their work, seven of them revealed to have received assistance from the government, NGOs, or INGOs. This support according to these organisations had been in the form of cash (funds), training, cleaning supplies, and medical supplies to support their operations.

Official registration is a requirement by the local law for CBOs' operation. The registration status of a CBO can influence whether a government or NGO/ INGO would wish to establish a relationship with it. The authors therefore asked about the registration status of the CBOs. All of them interviewed reported that they are officially registered to perform their respective functions. This report of increased registration compliance by CBOs in the study site could be linked to the needs of being formally recognised and gain the associated supports (Table 3).

### Discussion

The results of this study show the unique socio-demographic characteristics of respondents which have been shaped by their environment. For instance, Terpstra and Lindell suggest that gender indirectly influences the decision to respond to a hazard, and that women are more prone than men to identify risk and hazard-related traits (Terpstra and Lindell, 2013). It was also identified in a Serbian research study that males are viewed as being more ready for flooding and as having assurance in their capacity to handle it (Cvetković et al., 2018). In relation to these examples, in the slums of Portee and Rokupa there was no noticeable readiness to manage floods in the study site with respect to the greater proportion of female household heads among the interviewees. The existing socioeconomic and gender disparity in the country (Statistics Sierra Leone, 2019) may have contributed to this result. In terms of respondents' marital status, prior research in Accra, Ghana found a correlation between marital status and responding to floods in a protective rather than non-protective ways (Twerefou et al., 2020). Despite having more married people among interviewees in the study area did not indicate any flood protective behavior.

**Table 2.** Community-based organisations' role in flood management

| Name of CBO | Community | Clearing drainages, putting sandbags and other flood breakers | Organising community people and resources during flood emergencies | Help people during evacuation | Provide humanitarian relief items | Help rebuild destroyed homes and common assets | Problem identification | Community Sensitisation |
|---|---|---|---|---|---|---|---|---|
| Destiny sisters social club | Portee | Yes | Yes | No | No | Yes | No | Yes |
| Solar city organisation | Portee | Yes | No | No | No | No | No | Yes |
| Seaside rangers social club | Portee | Yes | No | No | Yes | Yes | No | No |
| Young stars' social club | Portee | Yes | Yes | No | No | Yes | No | No |
| Estate family social club | Portee | Yes | Yes | No | Yes | No | No | Yes |
| Community health workers' Organisation | Portee | Yes | Yes | No | Yes | No | Yes | Yes |
| Heaviest fashion social club | Rokupa | Yes | Yes | No | No | No | No | Yes |
| Sabenti Organisation | Rokupa | Yes | Yes | No | No | No | No | Yes |
| Moyen organisation | Rokupa | Yes | Yes | No | No | No | No | Yes |
| To me, to you Organisation | Rokupa | Yes | No | No | No | Yes | No | No |
| One family organisation | Rokupa | Yes | No | No | No | No | No | Yes |
| Fashion models' social club | Rokupa | Yes | No | No | No | Yes | No | Yes |
| **Total** | | **12** | **7** | **0** | **2** | **5** | **1** | **9** |

**Table 3.** Summary of CBOs and their support from external sources

| CBOs | Support received | | |
|---|---|---|---|
| One Family Organisation | Funding | | |
| To me to you, Organisation | Funding | Cleaning tools | |
| Moyen Organisation | Funding | Training | |
| Community Health Workers Organisation | Funding | Training | Medical items |
| Estate Family Social Club | Funding | Training | |
| Sea-Side Rangers Social Club | Funding | | |
| Destiny Sisters Social Club | Training | Funding | |

Research in Dar es Salaam, Tanzania shows that 77% of flood-prone urban slum households interviewed got support from a variety of sources, whereas only 23% reported receiving no support This stands in contrast to this study where household respondents claimed to have received little support (John, 2020). Only 10.8% of respondents who resided in underprivileged coastal urban households in Lagos reported receiving government assistance during floods (Adelekan, 2010), which is similar to the findings of this study.

The higher number of interviewed CBOs confirmed being supported by the government to carry out their operations represents an improvement over, when a previous study found CBOs disagreeing with the local government that it was not providing them with the necessary support (Macarthy et al., 2017).

It is detailed in the Community Resilience Theory that information and communication about a disaster form a solid foundation to reduce harm and build resilience and are vital to the development of economic resources, reducing risk and resource inequities, and attending to the community's areas of greatest social vulnerability (Norris et al., 2008). Therefore, information and communication about flooding mainly channelled by word of mouth in the study area can be considered inefficient for early warning and suggests a call for immediate attention.

### Guidelines for planning and research improvement

This paper put forward the following guidelines for consideration by the concerned authorities for improving planning and research in this subject.

A useful instrument, the draft Freetown structural plan (GOPA-CES, 2014), was built with solid recommendations for improved urban planning and housing, with considerations to reduce urban vulnerability to both anthropogenic and climate-related hazards and disasters, like flooding. However, the plan was guided by the outdated Local Government Act (2004). Therefore:

- It is needful for the plan to be finalised in consistency with the new Local Government Act (2022) to account for current happenings, potential climate change, and associated hazard scenarios.
- It is required that the finalisation team of the plan include an ideal representation of (community-chosen) women, men, the physically challenged, children, and religious and youth groups in slum households.
- These groups (community-selected representatives) can be further trained to obtain the capacity required to facilitate result-oriented involvement. This move, if considered, will allow for more participatory and inclusive decisions, as well as easier implementation.
- Any other local plans that will be developed in the future should be built on empirical evidence, be future-oriented, consider the least influential and sidelined community members, and be flexible to changes and revisions in anticipation of uncertainties.

*To improve and make future research result-oriented:*

- The perception of community-based readiness for flood management are necessary to be studied using primary data.
- Similarly, future research projects should explore intra- and interhousehold levels of vulnerability and preparations for current and possible impending flood events.
- Also, the relationships between household characteristics and the likelihood of using various flood management measures using correlation and other pertinent statistical tests is considered important by the authors.
- It is needful to conduct interdisciplinary research to determine the viability, socio-ecological and economic costs, and opportunities of transforming these slum settlements into contemporary flood-resilient buildings.
- Furthermore, future work should look at individuals' preferences and needs to permanently leave their present slum settlements.

### Conclusion

This study has revealed critical information on the readiness and response of households and CBOs to flooding in coastal slums in Freetown, Sierra Leone. The study's findings indicate that the majority of respondents learn about floods through word-of-mouth. This kind of information distribution is ineffective because it frequently falls short of getting to everyone who needs it in time for early preparation. This evidence explains the participants' widespread lack of flood knowledge and stands out as one of the main reasons for flood tragedies among them. It's important to note that the majority of household respondents reported having no one to turn to for assistance when floods occur.

This study is constrained by its disciplinary scope, as readiness has to do not only with planning and preparation with the available resources and information but also with the psychological state of willingness. Another limitation of the study is that it only investigated the household heads; this was done due to their responsibility and governing roles in their respective households, as well as the time and resources that would have been required to study individual members of each household. Although this work only intends to address issues within its scope, it lays a strong foundation that future broader projects can build to address interconnected subject concerns.

Similar to the Flood Rescue System developed by Hyoungseong et al. (2016), this study proposes the implementation of an interdisciplinary and interinstitutional initiative led by the Ministry of information and Communication, the national disaster management agency, and the Freetown City Council for the development of a "hazard situational assessment and report" smartphone application. The content of the application can include the forecasted weather situation and events of at least a week, a function to determine the location of the reporter and event or potential event, an automatic voice or text notification system, a function to upload a picture, and a brief report of the current or possible flooding situation. This joint institutional initiative should train representative groups of the most exposed households of the smartphone application use, and donate the device along with a durable mobile rechargeable (solar) power bank and free data access sim cards for the application's use. A follow-up training should be done at the start of every year's rainy season. During these follow-up periods, the repair of the smartphones and the update or upgrade of the proposed application should be looked at. We argue that the development and implementation of this idea will improve the early warning system, enable well-informed decisions and early preparation, reduce risk, and boost the flood resilience of the study area.

Finally, we encourage CBOs to strengthen inter-community and organisational learning, feedback, and support systems and adopt a "no wait on the government principle" to streamline the necessary flood response and management actions within their capabilities in a way that losses and damage can still be significantly minimised even with a late arrival, inadequate or no external support.

**Open peer review.** To view the open peer review materials for this article, please visit http://doi.org/10.1017/cft.2023.29.

**Acknowledgement.** The authors appreciate the people of Portee and Rokupa for their cooperation in the study. We further acknowledge the community-based expert for his support during the fieldwork.

**Author contribution.** Bashiru Turay (BT) conceived the presented idea; Bashiru Turay and Sheku Gbetuwa, designed the research instrument and methodology; Alieu Turay and Sheku Gbetuwa provided additional relevant literature and recommendations; Bashiru Turay prepared the final draft. All authors contributed to the fieldwork, and reviewed and approved the final manuscript submission to this journal for publication.

**Financial support.** This research did not receive any specific grant from funding agencies in the public, commercial, or not-for-profit sectors.

**Competing interest.** The authors declare that they have no known competing financial interests or personal relationships that could have appeared to influence the work reported in this paper.

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
