## [Reviewer Report]

Review

The manuscript titled “Households’ Readiness and Community-based Organizations’ Response to Flooding: the case of Freetown City’s coastal area” conducted in Portee and Rukupa, a coastal slum in Freetown, Sierra Leone, aims to explore household readiness and community-based organizations’ responses to flooding. This is an interesting investigation, however, carries some gaps for improvement, that I think doesn’t allow me to accept it in its present form. I am forwarding the comments section by section, and also request the authors that next time submit the revised version with line numbers so that to make it easier for you to follow the comments point by point where improvement is necessary.

Title:

1. I think the title can be improved by including the word, “Flood Management” because the manuscript is all about management.

Abstract:

2. The abstract should have information about the techniques used in the study and the sample size on which the results are based.

3. There should be some recommendations regarding the results mentioned in the text in addition to the future recommendations

Introduction:

4. I suggest the authors do restructure the introduction section on the flowing suggestions:

i. Broadly start from the floods impacts, and prove that how it’s an important issue for you.

ii. What are the socio-economic losses of the 2020 floods and if more historical data is available?

iii. Link your flood impacts with poor disaster management.

iv. What does draft Freetown structural plan 2013–2028 say? How it can help in better flood management.

v. I didn’t see on which theory this study is based i.e, Social-ecological systems theory, Risk and vulnerability theory, Community resilience theory. I suggest the author to present a theorical/conceptual model, that can clearly show the whole research.

5. Methodology

i. If you have the household size, why the study didn’t use the survey? It is not justified.

ii. Why you have used the observation as a tool of your data collection.

iii. Only selected 204 households fulfilled the criteria.

iv. Why 12 representatives of community-based organizations were interviewed using a snowball sampling technique? Any justification.

v. What is the validity and reliability of this tool (The Kobo toolbox)?

vi. State the inclusion and exclusion criteria in this section.

vii. Provide the methods of data analysis in a separate section

6. Results and discussion

i. Present the households characteristics in table.

ii. Separate the results and discussion section so that one can easily see and follow the results and their implications.

iii. “Women are more prone than men to identify risk and hazard related trait” how can you say this, such statement are many in the results section, please support with solid justification from your results and data analysis.

iv. The results should be like the way you presented in Table 1.

7. Conclusion

i. I didn’t see the discussion about Sustainable Development gaols, Sendai Framework on Disaster Risk Reduction, however, included in the conclusion. The conclusion should be very specific based on your findings, and more focused on implications. Summarise the conclusion section as you can, this is very lengthy looks like the whole discussion section.

---

## [Editor Report]

Dear Dr. Gbetuwa,

I have received one detailed review of the manuscript you submitted to Coastal Futures and have read your paper myself. The reviewer provided detailed comments and some practical suggestions for change. I am recommending that you consider the comments from the reviewer and re-submit after a major revision.

---

## [Editor Report]

Dear Mr. Gbetuwa,

Following a second review of the manuscript I am satisfied to recommend that the manuscript can be considered for publication, but only following minor revisions. It is however important to note that some changes may still require some rethinking. Overall, the response letter from the authors was adequate based on the first round of reviews. The revised version of the manuscript was easier to read and more focused. I hope that your next response will further clarify some additional issues.

Please note that I have provided some language editing comments and suggestions in the attached manuscript. I also provide a few questions that need to be clarified in the manuscript. Please respond carefully to the comments in the annotated manuscript and in particular respond to the following:

• The 2nd paragraph of the Conclusion seems to make more sense as a subsection of the Discussion section.

• It is not clear how 3rd paragraph of the conclusion section makes sense as a conclusion of your study? Did your respondents ask for such a tool? Please make the case for how this tool, with the features and details you give is connected to the study that this paper reports on? Has a similar tool been implemented elsewhere successfully?

Best Regards,

Peter Ruggiero

---

## [Editor Report]

Dear Mr. Gbetuwa,

Thanks very much for your revised manuscript and detailed response to the last round of reviews. I am happy to now recommend the manuscript for publication.

Best,

Peter Ruggiero